# Luteolin Attenuates APEC-Induced Oxidative Stress and Inflammation via Inhibiting the HMGB1/TLR4/NF-κB Signal Axis in the Ileum of Chicks

**DOI:** 10.3390/ani13010083

**Published:** 2022-12-26

**Authors:** Zhanyou Cao, Chenghong Xing, Xinyi Cheng, Junrong Luo, Ruiming Hu, Huabin Cao, Xiaoquan Guo, Fan Yang, Yu Zhuang, Guoliang Hu

**Affiliations:** College of Animal Science and Technology, Jiangxi Agricultural University, Nanchang 330045, China

**Keywords:** luteolin, intestinal injure, APEC, HMGB1, NF-κB, chick

## Abstract

**Simple Summary:**

Avian colibacillosis is one of the major causes of animal death and egg-production decline, which has caused economic losses to the poultry industry. When encountering avian colibacillosis, many poultry farms tend to use antibiotics to reduce losses. Additionally, the utilization of antimicrobials is also disputable. Our study shows that luteolin can alleviate the inflammation and oxidative stress caused by *Escherichia coli* in the ilea of chicks and may be used as a substitute for antibiotics to control avian colibacillosis.

**Abstract:**

Avian pathogenic *E. coli* (APEC) is typically the cause of avian colibacillosis, which can result in oxidative stress, inflammation, and intestinal damage (APEC)**.** Luteolin, in the form of glycosylation flavone, has potent anti-inflammatory and anti-oxidative properties. However, its effects on APEC-induced intestinal oxidative stress and NF-κB-mediated inflammation in chicks remains poorly understood. After hatching, one-day-old chicks were stochastically assigned to four groups: a control group (basic diet), an *E. coli* group (basic diet) and L10 and L20 groups (with a dry matter of luteolin diet 10 mg/kg and 20 mg/kg, respectively), with fifteen chicks in each group and one repeat per group. They were pretreated for thirteen days. The body weight, mortality, histopathological changes in the ileum, antioxidant status, and the mRNA and protein-expression levels of factors associated with the HMGB1/TLR4/NF-κB signal axis of the chicks were measured. The results showed that luteolin treatment decreased the mRNA and protein-expression level of the related factors of HMGB1/TLR4/NF-κB signal axis in the ileum, reduced inflammation, increased antioxidant enzyme activity, and reduced intestinal injury. Collectively, luteolin alleviated APEC-induced intestinal damage by means of hindering the HMGB1/TLR4/NF-κB signal axis, which suggests that luteolin could be a good method for the prevention and treatment of avian colibacillosis.

## 1. Introduction

As one of the main causes of animal fatalities and egg-production reductions, avian colibacillosis causes a significant financial loss to the poultry business. APEC is the main cause of colibacillosis in poultry farms and can cause extra-intestinal infections, including emphysema, pneumonia [1] and intestinal infections (diarrhea and enteritis) [2]. Early in life, the intestinal immune system of chicks is immature, making them more susceptible to APEC infection, and the ileum is an important absorption site that is especially vital for newly hatched chicks [3]. One of the causes of endogenous free radicals is bacterial contamination, which actuates immune cells and actuates inflammation. Organisms secrete different pro-inflammatory cytokines to mediate the immune system’s protection against specific pathogens, but excessive inflammation can cause oxidative stress and damage to normal tissues [4]. High-mobility group box 1 (HMGB1), a damage-associated molecular-pattern molecule (DAMP), may be released into the extracellular environment and can connect with certain receptors, primarily toll-like receptors (TLRs) on target cells, initiating an inflammatory response [5,6]. TLR4 can recognize and bind to extracellular HMGB1, activating the myeloid differentiation primary response protein 88 (MyD88) pathway and the NF-κB signaling pathway, thereby triggering a cascade reaction [7,8]. A combined application of antibiotics is the main method used to prevent and treat intestinal diseases caused by APEC in worldwide poultry farms [9]. However, excessive use of antibiotics in disease prevention can lead to the widespread spread of genes with antimicrobial properties by promoting the selection of antibiotic-resistant microorganisms in animals [10,11,12]. Therefore, there is an increasing interest in replacing antibiotics with natural products such as probiotics [13,14] or medicinal plant extracts [15,16] to resist pathogen invasion and improve poultry-growth performance.

Luteolin (3,4,5,7-tetrahydroxyflavone), in the state of glycosylation flavone, is mainly present in medicinal plants and vegetables such as carrots, chamomile tea, and honeysuckle [17]. It has powerful anti-inflammatory and anti-oxidation effects; thus, luteolin has received extensive attention [18]. Several studies have shown that plant extracts such as baicalin, turmeric oleoresin, and capsicum oleoresin can improve the oxidative stress and inflammatory response evocable by APEC [19,20]. Luteolin inhibited nuclear factor kappa-B (NF-B)-mediated inflammation and triggered nuclear factor E2-related factor 2 (Nrf2)-mediated antioxidant responses can protect against diabetic cardiomyopathy, as demonstrated by Li et al. [21]. Additionally, luteolin has an anti-inflammatory effect via the downregulation of phosphorylated signal transducers, the activation of transcription (p-STAT) 3 and the upregulation of p-STAT6 [22]. However, there are currently no specific testimonies that luteolin can improve APEC-induced intestinal damage by inhibiting intestinal oxidative stress and inflammation. Additionally, it has been demonstrated that flavonoid compounds such baicalin, quercetin, apigenin, and baicalein can interfere with quorum sensing, biofilm formation, and the expression of virulent genes to lessen the pathogenicity of APEC [23,24,25]. It was also reported that luteolin reduced the expression level of the HMGB1/NF-κB signal axis in the intestinal tracts of mice, thereby alleviating dextran sodium sulfate (DSS)- incited colitis [26]. However, it is not clear whether luteolin can alleviate APEC-induced intestinal injury through the HMGB1/TLR4/NF-κB signal axis. The current study intends to examine the protective effects of luteolin on inflammation and oxidative stress caused by APEC in the ilea of chicks, as well as to further explore its underlying mechanism.

## 2. Materials and Methods

### 2.1. Experimental Animal and Treatments

The Ethics Committee of Jiangxi Agricultural University approved the use of laboratory animals and the associated protocols (No. JXAULL-2020-30), and all experimental methods were carried out in accordance with the “Guidelines for the Management and Use of Laboratory Animals” established by Jiangxi Agricultural University’s Laboratory Animal Management and Use Committee. *E. coli* APAP-O78 and luteolin were provided by the China Veterinary Drug Administration (# cvcc 1418) and Sigma−Aldrich (the US, Product No. 72511), respectively. Hy-line-Brown-variety chicks on the first day of hatching were stochastically separated into four groups with fifteen chicks in each group. The experimental groups included the control group, the Escherichia coli group (*E. coli* group), the 10 mg/kg luteolin group (L10 group), and the 20 mg/kg luteolin group (L20 group). The composition of their basic diet is shown in the Table 1. The control group and *E. coli* group were given the basal diets, while the L10 and L20 groups received the basal diet containing luteolin (10 mg/kg, 20 mg/kg). Luria–Bertani (LB) broth was used to grow the *E. coli* culture at 37 °C. The bacteria’s colony-forming units (CFU) were then calculated using the drawing board counting method. On the fourteenth day, chicks in the *E. coli*, L10, and L20 groups were injected intramuscularly with a single injection of a 0.2 mL/animal dose of Escherichia coli liquid (3.39 × 10^9^ CFU/mL), and the control group was intramuscularly injected with the same dose of normal saline. On the sixteenth day, six randomly selected chicks from each group were anesthetized by intravenous overdose of pentobarbital sodium. After sacrifice, ilea were collected on ice and the samples were stored at −80 °C.

### 2.2. Body Weight and the Mortality of Chicks

The initial body weight (IBW) (one-day-old chicks), body weight gain (BWG) (final weight–initial weight), final body weight (FBW) (thirteen-day-old chicks), average daily gain (ADG) (body weight gain/the number of days), and the fatality rate of the chicks ((number of dead chicks/total number of chicks in each group) × 100%) were calculated on the sixteenth day.

### 2.3. Hematoxylin-Eosin (H&E) Staining

The ileal tissues were removed on the sixteenth day of the experiment, washed with ice-cold saline, and fixed in a 4% formaldehyde solution. The samples were then embedded in paraffin, cut into 5 micron sections, and stained with hematoxylin and eosin. The pathological changes of the ilea were observed under a light microscope (Olympus, Tokyo, Japan).

### 2.4. Immunofluorescence (IF) Staining

The ileal tissues of chicks were removed at an appropriate time and fixed with liquid nitrogen. Myeloperoxidase (MPO) was incubated on the sections for 12 h at 4 °C after being diluted to a concentration of 1:200. Sections were treated with goat anti-rabbit or anti-mouse immunoglobulin (Ig) G that had been fluorescein (FITC)-tagged after being washed with phosphate buffer (PBS) (1:1000, diluted). The slices that had been cleaned with PBS were stained with 4′,6-diamino-2-phenylindole (DAPI). Finally, a fluorescence microscope (Nikon Eclipse C1, Tokyo, Japan) was used to view and photograph the slices’ fluorescence.

### 2.5. Determination of Antioxidant Activities

According to the manufacturer’s operating procedures, the activities of antioxidant enzymes glutathione peroxidase (GSH-Px), catalase (CAT), superoxide dismutase (SOD), and the concentration of malondialdehyde (MDA) in the chicks’ ileal samples were measured using commercial kits (Nanjing Jiancheng Bioengineering Research Institute, Nanjing, China).

### 2.6. Real-Time PCR (RT-PCR)

TransZol Up (TransGen Biotech, Beijing, China) was used to extract the total RNA, and EasyScript^®^ One-Step gDNA removal and cDNA Synthesis SuperMix (TransGen, Beijing, China) were used to reverse-transcribe the total RNA into cDNA in order to assess the mRNA level of the target genes. Utilizing the ChamQ SYBR qPCR Master Mix, RT-PCR was carried out (Vazyme, Nanjing, China). Primer Premier Software was used to create the primers for HMGB1, MYD88, TLR4, NF-B, interleukin (IL)-1β, IL-6, IL-8, IL-10, TNF-α, IL-13, heme oxygenase-1 (HO-1), SOD1, SOD2, CAT, glutamate cysteine ligase modifier subunit (GCLM), and glutathione peroxi-dase-1 (GPX1) (Premier Bio soft International, San Francisco, CA, USA). Utilizing the Quant Studio 7 Flex RT-PCR machine, the reaction was carried out (ABI 7900HT Applied Biosystems, Applied Biosystems, Foster City, CA, USA). Table 2 presents the primer sequences. Using GAPDH as an internal reference gene, the relative changes in gene mRNA levels were assessed using the 2^−ΔΔCT^ technique.

### 2.7. Western Blotting

The samples were homogenized at 4 °C in a Radio Immunoprecipitation Assay (RIPA) buffer lysis (Solarbio Biotecnology Beijing, China), which included the protease inhibitor phenylmethyl sulfonyl fluoride (PMSF) (Beyotime, Shanghai, China). The protein content of the chicks’ ileal samples was determined using a bicinchoninic acid (BCA) kit (Solarbio Biotecnology Beijing, China). Samples were further diluted, and an SDS-PAGE loading buffer was added and boiled for 10 min (100 °C). A protein sample (30 μg) was added to SDS- polyacrylamide denatured gel and then transferred to a polyvinylidene fluoride (PVDF) membrane after electrophoresis. The primary antibodies were anti-HMGB1 (1:500; Wanleibio, Shenyang, China), anti-p65 (1:1000; Bimake, Shanghai, China), anti-p-p65 (1:500; BioGot, Nanjing, China), and anti-TLR4 (1:500; Wanleibio, Shenyang, China), and the GAPDH bands were used as controls and quantified standardized by Image Lab and Image J software.

### 2.8. Statistical Analysis

Microsoft Excel 2020 was used to sort the data collected throughout the trial. For statistical data analysis, the student’s t-test with two tails or the one-way analysis of variance (ANOVA) and Tukey’s post-hoc test were used. The mean and standard deviation (SD) were used to express data. Finally, the test results were plotted using the Prism 8.0 program. At *p* < 0.05, the difference was deemed significant.

## 3. Results

### 3.1. Effects of Luteolin on Body Weight, Mortality Rates and Diarrhea Rate of Chicks

The body-weight changes of chicks are presented in Table 3 and Figure 1A. Between the control group and theL10 and L20 groups, there was no change in IBW (*p* > 0.05). The FBW, BWG, and ADG of the L10 and L20 groups were clearly higher than those of the control group (*p* < 0.05). After *E. coli* injection, the mortality rate and diarrhea rate of the L10 and L20 groups was inferior to that in *E. coli* group showing in Figure 1B. The above effects of luteolin on chicks increased in a dose-dependent manner.

### 3.2. Histopathological Analysis of Ileum Tissues

The ilea were collected for H&E staining (Figure 2). As the picture shows, ileal villi were neatly arranged, the cells were clearly defined, and the glands were complete in the control group. There were typical inflammatory changes, such as shortened ileal villi, incomplete glands, extensive infiltration of inflammatory cells, lymphoid follicle emptying, and hyperemia (as shown by the arrow) in the *E. coli* group. In contrast, in the L10 and L20 groups, there were no or few pathological changes such as infiltration and diffusion of inflammatory cells.

### 3.3. Immunofluorescence (IF) Staining

Figure 3A,B demonstrates that the MPO content of the *E. coli* group significantly increased (*p* < 0.05) when compared to the control group. The expression of MPO was, however, noticeably lower in the L10 and L20 groups than in the *E. coli* group (*p* < 0.05).

### 3.4. Luteolin Attenuates the Level of Oxidative Stress in Chick Ileum

To determine the effects of luteolin on the level of oxidative stress in the ilea, we measured SOD, GSH-Px, CAT activities, and MDA content using the kits (Figure 4A–D). The activities of SOD and GSH-Px in the control group were arrested higher than those in the *E. coli* group (*p* < 0.05), and CAT activity slightly exceeded that in the *E. coli* group. The activities of SOD and CAT in the luteolin group were noteworthily higher than those in the *E. coli* group (*p* < 0.05), but it did exhibit an increasing trend. The MDA concentration in the L20 group was substantially lower than that in the *E. coli* group (*p* < 0.05).

Additionally, we measured the mRNA expression levels of antioxidant genes HO-1, SOD1, SOD2, CAT, NQO1, GCLM, and GPX1, which are shown in Figure 4E. The mRNA levels of HO-1, SOD2, CAT, NQO1, GCLM, and GPX1 in the control group conspicuously exceeded those in the *E. coli* group (*p* < 0.05). When compared to the *E. coli* group, the mRNA levels of HO-1, SOD1, SOD2, CAT, NQO1, GCLM, and GPX1 of the L10 group were noticeably increased (*p* < 0.05), and the SOD1, SOD2, GCLM, and GPX1 mRNA levels in the L20 group were remarkably upregulated (*p* < 0.05). Furthermore, there was no obvious difference in the mRNA expression levels of CAT, NQO1, and HO-1 between the L20 group and the *E. coli* group (*p* > 0.05). Apparently, luteolin can mitigate the oxidative stress induced by APEC to a certain extent.

### 3.5. Effect of Luteolin on the mRNA Expression Levels of Cytokine

We detected the mRNA expression levels of IL-10, IL-8, IL-1β, IL-6, TNF-α and IL-13, and the results are shown in Figure 5. The IL-1β, IL-6, IL-8, IL-10, and IL-13 mRNA expression levels in the control group were noticeably lower than those in the *E. coli* group (*p* < 0.05). The mRNA expression levels of these cytokines in the L10 and L20 groups were noticeably lower than those in the *E. coli* group (*p* < 0.05).

### 3.6. Effects of Luteolin on HMGB1/TLR4/NF-κB Signaling Pathway

Figure 6A demonstrates that the TLR4, NF-κB, HMGB1, and MYD88 mRNA expression levels in the *E. coli* group were noticeably greater than those in the control group (*p* < 0.05). The mRNA expression levels of HMGB1, MYD88, and NF-κB in the luteolin group were noticeably lower than those in the *E. coli* group (*p* < 0.05). The TLR4 mRNA expression level in the L10 group was significantly lower (*p* < 0.05) compared to the *E. coli* group, whereas it was only marginally lower (*p* > 0.05) in the L20 group.

The protein expression levels of HMGB1, p-p65, TLR4, and p65 are shown in Figure 6B–G. In comparison to the *E. coli* group, the protein expression levels of HMGB1, TLR4, and p-p65 in the control group were noticeably lower (*p* < 0.05). The protein expression levels of HMGB1 and p-p65 in the L10 and L20 groups significantly decreased (*p* < 0.05) compared to the *E. coli* group. Compared to the *E. coli* group, the protein expression level of TLR4 was significantly lower in the L20 group (*p* < 0.05) and marginally lower in the L10 group (*p* > 0.05).

## 4. Discussion

Avian colibacillosis arising from APEC is a serious intestinal disease in poultry production and can result in decreased production performance and increased mortality rates [27]. This may be due to the imperfect immune system development of newly hatched chicks that are highly sensitive to various stressors, including oxidative stress and inflammation caused by bacterial pathogens. According to studies, luteolin has powerful anti-inflammatory and anti-oxidative stress actions [21,26]. In order to understand the preventive effect of luteolin on APEC-induced intestinal damage and its potential mechanism, we investigated the markers of oxidative stress and inflammation in this study. The findings demonstrated that luteolin, via the HMGB1/TLR4/NF-κB signal axis, may enhance the development of chicks, lower the degree of ileal oxidative stress, and ameliorate APEC-induced intestinal inflammation.

A healthy and intact intestinal morphology is important for the absorption of nutrients, which may have a greater impact on the growth of newly hatched chicks [28]. Additionally, MPO is a quantitative measure of neutrophil activity in the small intestine, which might increase intestinal inflammation by producing oxygen free radicals in the intestinal barrier [29,30]. Our research shows that APEC causes the shortening of ileal villi, extensive infiltration of inflammatory cells, bleeding, and increases the amount of MPO-caused ileal inflammation and even tissue injury. However, luteolin can alleviate these effects. During APEC infection, excessive inflammation leads to mitochondrial dysfunction and oxidative stress [31]. Antioxidants such as SOD, GSH-Px, and CAT can affect cellular capacity to scavenge free radicals and regulate oxidative stress. Since MDA is a byproduct of lipid peroxidation, its concentration indicates the degree of oxidative stress [32,33]. In this study, the mRNA levels of antioxidant genes (HO-1, SOD1, SOD2, CAT, NQO1, GCLM, and GPX1) in the luteolin treatment groups were increased, the activities of GSH-Px, CAT, and SOD were upregulated, and MDA levels in the ilea were lower than that in the *E. coli* group. Studies have shown that luteolin can antagonize the decrease of CAT, SOD, and GSH-Px activities and increase the MDA concentration in rat kidneys induced by lead acetate (PbAc), which is consistent with our results [34].

HMGB1 is a late-stage inflammatory factor, which is released after the release peak of early inflammatory factors involved in the inflammatory response. It can trigger the inflammatory response, and even result in uncontrolled inflammation [35]. During infection or injury, HMGB1 also functions outside the nucleus [36]. A cellular inflammatory response is triggered when extracellularly released HMGB1 binds to target cells’ TLR4 receptors and activates the downstream MyD88 and NF-κB signaling pathways. Additionally, the NF-κB pathway’s activation can increase the expression of cytokines that cause inflammation, such as IL-6, IL-1β, and TNF-α, etc. [37,38]. Inflammatory-reduction agents’ pro-inflammatory cytokines such as IL-1β and IL-6 can be prevented from being released by IL-10 and IL-13. When the inflammatory level rises, the body may control the inflammatory response by regulating the production of anti-inflammatory cytokines [39,40]. Zuo et al. showed that luteolin relieves DSS-induced colitis in mice through the HMGB1/TLR4/NF-κB signal axis [26]. According to our data, the mRNA levels of pro-inflammatory cytokines (IL-1β, IL-6, IL-8, and TNF-α) and anti-inflammatory cytokines (IL-13 and IL-10) were upregulated in the *E. coli* group, whereas the treatment groups experienced the opposite impact. These results indicate that luteolin could relieve inflammation caused by APEC in the ilea of the chicks by inhibiting the HMGB1/TLR4/NF-κB-signal axis. In addition, studies have shown that luteolin has anti-inflammatory and antioxidant functions, and this effect is enhanced in a dose-dependent manner [41,42,43]. Our results, which were consistent with previous studies, showed that luteolin could increase mRNA expression levels of antioxidant genes (GCLM, GPX1, SOD1, and SOD2) and inhibit mRNA expression levels of inflammatory cytokines (IL-1β, IL-6, TNF-α, and IL-13). Additionally, luteolin can promote the growth of chicks and reduce the diarrhea rate and mortality rate induced by APEC. Similarly, the above-mentioned effects on chicks of having luteolin in their diet showed a dose-dependent effect.

Several studies on anti-inflammatory effects have been reported [44,45], but there is a lack of scientific knowledge about the potential impacts of luteolin on intestinal inflammation, especially the potential molecular mechanisms of luteolin exerting its anti-inflammatory effects. This study describes the anti-oxidative stress and anti-inflammatory effects of luteolin and introduces a novel signaling axis involved in APEC-induced inflammation, which is beneficial to gaining a better understand of anti-inflammatory mechanism of luteolin. There is a need for more discussion on luteolin inhibiting inflammation through the HMGB1/TLR4/NF-κB signal axis. However, in this study we only investigated Hy-Line Brown laying chicks. More research is needed to further clarify whether this effect is also observed in chickens of other ages and breeds. In addition, the potential benefits of plant extracts may be different due to the great differences in the components of plant extracts [46]. At present, the research on the pathway and biological function of active components on target organs is still in the stage of speculation, and there is no strong scientific basis. Future research should focus on the biological function mechanism of the Chinese herbal medicine extract, clarify its mechanism of action, and lay a theoretical foundation for the rational utilization of plant-extraction products to promote wider application.

## 5. Conclusions

In summary, the results indicate that luteolin relieves oxidative stress, inflammation, and ileal tissue damage caused by APEC in chicks. This study describes the HMGB1/TLR4/NF-κB signal axis, which is a new signaling axis that may be involved in APEC-induced inflammation and helps us to better understand the mechanism of luteolin in treating inflammatory diseases aroused by bacteria.

## Figures and Tables

**Figure 1 animals-13-00083-f001:**
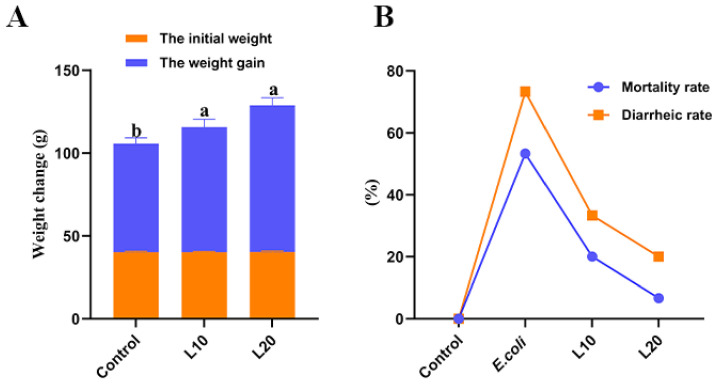
(**A**) Body-weight changes of chicks fed luteolin for 13 days. (**B**) The mortality rates (%) and diarrhea rates (%) of chicks in the control group, *E. coli* group, L10 (10 mg/kg luteolin) group, and L20 (20 mg/kg luteolin) group after *E. coli* injection. The data are expressed as mean ± SD. A statistical difference (*p* < 0.05) is indicated using different letters.

**Figure 2 animals-13-00083-f002:**
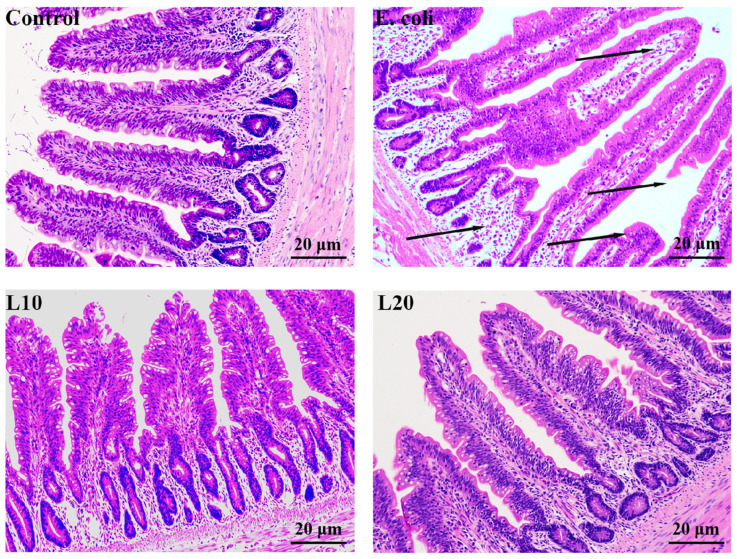
Effect of luteolin on pathological changes of the ileum tissues in chicks. Hematoxylin and eosin (HE) staining are shown (200×). The black arrows manifest the location of the lesions.

**Figure 3 animals-13-00083-f003:**
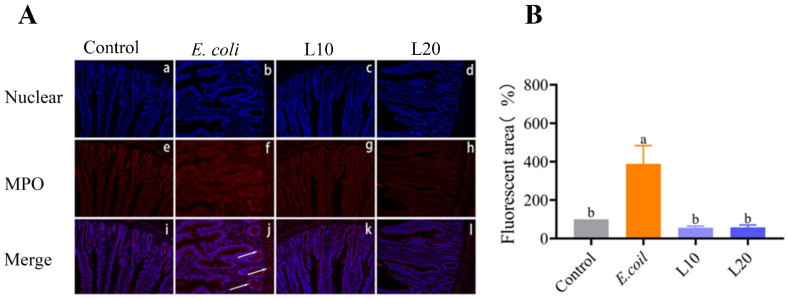
(**A**) The nuclear staining exhibited blue fluorescence in the photos (a–d), the MPO staining showed red fluorescence in the images (e–h), and the confluent fluorescence signals were displayed as merged images (i–l). The elevated signals in the *E. coli* group were shown by arrows. (**B**) The level of MPO expression. The data is presented as mean ± SD. A statistical difference (*p* < 0.05) is indicated using different letters.

**Figure 4 animals-13-00083-f004:**
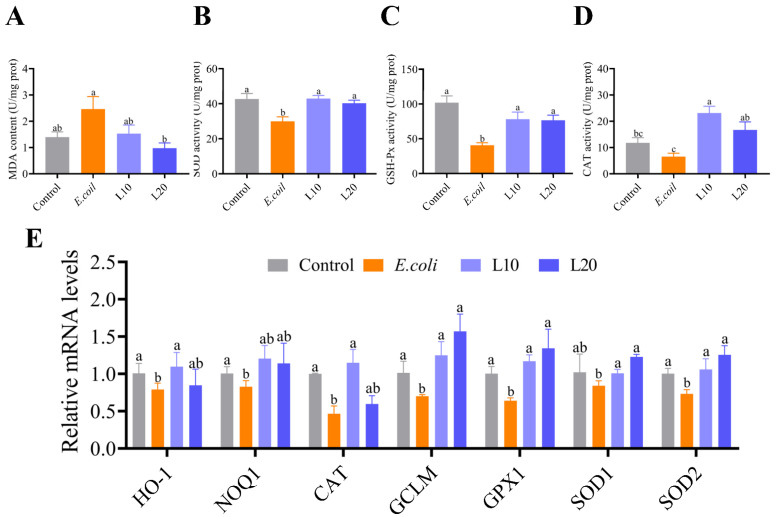
Effects of luteolin on oxidative stress in chicken ileum tissue. (**A**) MDA content, (**B**) SOD activity, (**C**) GSH-Px activity, (**D**) CAT activity, and (**E**) The mRNA expression levels related to antioxidant genes. The data are expressed as mean ± SD. A statistical difference (*p* < 0.05) demonstrated using different letters.

**Figure 5 animals-13-00083-f005:**
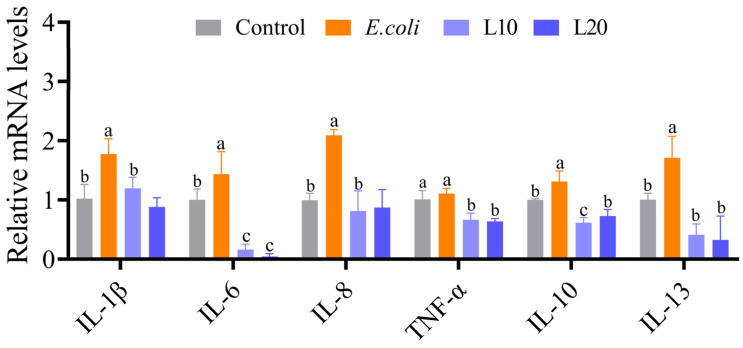
Luteolin treatment significantly improved the expression of *E. coli*-induced inflammatory cytokines. The effect of luteolin on the mRNA levels of inflammatory genes in each group. The data are expressed as mean ± SD. A statistical difference (*p* < 0.05) is shown by different letters.

**Figure 6 animals-13-00083-f006:**
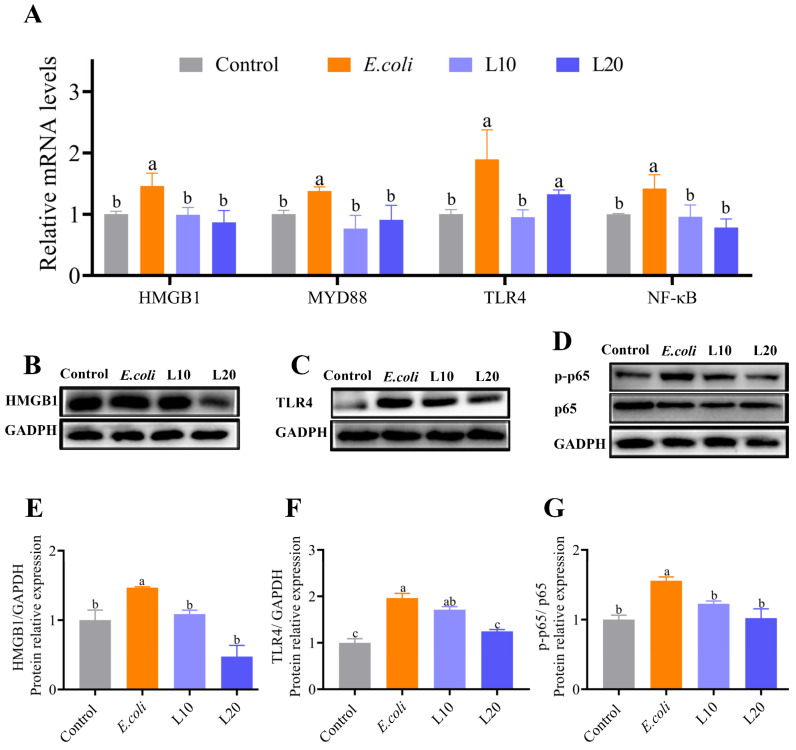
The effects of luteolin on HMGB1/TLR/NF-κB pathway-related genes and protein levels. (**A**) Changes in the mRNA levels of HMGB1/TLR4/NF-κB pathway-related genes in the control, *E. coli*, L10, and L20 groups, (**B**–**G**) The protein expression levels of HMGB1 (25 kDa), TLR4 (95 kDa), p-p65 (65 kDa), and p65 (65 kDa). The data are expressed as mean ± SD. A statistical difference (*p* < 0.05) is shown by different letters.

**Table 1 animals-13-00083-t001:** The elements and amounts of nutrients in the basic diet for chicks.

Ingredient	Content (%)
Corn	66.00
Soybean meal	29.00
Premix ^a^	5.00
Total	100.00
DL-Methionine %	0.26
Metabolic Energy (MJ/kg)	11.92
Lysine %	0.88
Crude protein %	17.61

^a^ Addition per kilogram of premix: DL-Methionine: 300 g; Sodium chloride: 2000 g; Calcium hydrogen phosphate: 6000 g; Polysaccharide vitamin: 100 g; Mountain flour: 13,000 g; Zinc: 87 g; Iron: 67 g; Cobalt: 94 g; Copper: 8 g; Potassium Iodide: 8.75 g; Sodium selenite: 11.25 g. All values were measured except for Metabolic Energy. Metabolic Energy was a calculated value.

**Table 2 animals-13-00083-t002:** Sequence of the gene primers and the GenBank accession number.

Gene Name	Accession Number	Primer Sequences (5′ to 3′)
GAPDH	NM_204305.1	Forward: AGTCGGAGTCAACGGATTTGG Reverse: GGTCAACATCGCCACCTACA
HMGB1	NM_204902	Forward: CGACTCTGACGCGGAAAATC Reverse: CGGCAGGTTTGCACAAAGAA
MYD88	NM_001030962.4	Forward: TTGACTTCTGCATGGGTCCT Reverse: TTGCTCCACAGTCACCAGAT
TLR4	NM_001030693.1	Forward: CATACAAGCCACTCCAAGCC Reverse: AGGATTTCCAGGGCTGAGTC
NF-κ B	NM_205129.1	Forward: GTGTGAAGAAACGGGAACTG Reverse: GGCACGGTTGTCATAGATGG
IL-1β	NM_204524.1	Forward: GGTCAACATCGCCACCTACA Reverse: CATACGAGATGGAAACCAGCAA
IL-6	NM_205498.1	Forward: AAATCCCTCCTCGCCAATCT Reverse: CCCTCACGGTCTTCTCCATAAA
IL-8	NM_205498.1	Forward: GCAAGGTAGGACGCTGGTAA Reverse: GCGTCAGCTTCACATCTTGA
IL-4	NM_001007079.1	Forward: AATTGTTTGGGAGAGCCAGCA Reverse: ATTCAGGAGCTGACGCATGTT
IL-10	NM_205129.1	Forward: GGGAGCTGAGGGTGAAGTTT Reverse: TCTGTGTAGAAGCGCAGCAT
TNF-α	NM_204608.1	Forward: CAGATGGGAAGGGAATGAAC Reverse: CACACGACAGCCAAGTCAAC
IL-13	NM_001007085.2	Forward: TCAAGGATCGGAAGCTGTCA Reverse: GTCCTTCTTGCAGTCGGTCA
HO-1	NM_205344.1	Forward: ACAACGCTGAAAGCATGTCC Reverse: GGATGCTTCTTGCCAACGAC
SOD1	NM_205064.1	Forward: CCAAAAGATGCAGATAGGCACG Reverse: GCAGTGTGGTCCGGTAAGAG
SOD2	NM_204211.1	Forward: TGGGGGTGGCTTGGGTATAA Reverse: CAGCAATGGAATGAGACCTGTT
CAT	NM_001031215.2	Forward: AGCTTGCAAAATGGCTGACG Reverse: ATAGCCAAAGGCACCTGCTC
NQO1	NM_001277620.1	Forward: CGCACCCTGAGAAAACCTCT Reverse: ACTGCAGTGGGAACTGGAAG
GCLM	NM_001007953.1	Forward: CGTGTGCTGAGTCACGGTGT Reverse: TCCAACAATGAAAAGTTTTGCCGA
GPX1	NM_001277853.2	Forward: CCAATTCGGGCACCAGGAGAA Reverse: GGTGCGGGCTTTCCTTTACT

**Table 3 animals-13-00083-t003:** The effect of luteolin-supplemented meals on chick development performance.

Trait Studied	Groups		
	Control	L10	L20
IBW (g)	38.37 ± 2.05	38.41 ± 2.34	38.48 ± 2.40
FBW (g)	105.13 ± 11.86 ^a^	125.87 ± 9.67 ^b^	134.53 ± 9.05 ^b^
BWG (g)	67.66 ±12.31 ^a^	87.56 ± 9.95 ^b^	96.05 ± 9.55 ^b^
ADG (g)	5.20 ± 0.94 ^a^	6.73± 0.76 ^b^	7.38± 0.73 ^b^

In the same index, different small letters indicated significant difference (*p* < 0.05); the same or no letter indicated no significant difference (*p* > 0.05). Number of observations (*n* = 15) of chicks in each group. The data were expressed as the mean ± SD.

## Data Availability

Not applicable.

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
