# Peer review of "Luteolin Attenuates APEC-Induced Oxidative Stress and Inflammation via Inhibiting the HMGB1/TLR4/NF-κB Signal Axis in the Ileum of Chicks"

_animals, 2022, doi:10.3390/ani13010083_

Round 1
Reviewer 1 Report
This manuscript entitled "Luteolin attenuates APEC-induced oxidative stress and inflam-2 mation via inhibiting the HMGB1/TLR4/NF-κB signal axis in 3 the ileum of chicks" mainly describes the luteolin can be a good method for the prevention and treatment of avian colibacillosis. The structure of the article is relatively complete, but it still needs some modifications, as follows:
General comments:
- The expression of the article in English needs further improving.
- The article mentions two ratios of Luteolin addition, with no explanation for the different results caused by the different ratios, nor does it suggest a better ratio or range of addition.
Specific comments:
Introduction
Line 63-82 Please regroup the two paragraphs to make the logic clearer and improve the second paragraph of the introduction.
Materials and methods
Please add the criteria for animal slaughter in this section.
Line 89 Please add which of the values in Table 1 are measured and which are calculated.
Results
Line 156-157 In section 3.1, this section of the textual narrative needs to state significance, as well as insignificance.
Line 163-164 "(B) The mortality rates (%) 163 and diarrhea rate (%) of chicks in control group". Please note that the mortality rates in the notes are consistent with those in the legend.
Line 235-239 Please explain why L10 and L20 have each had these effects.
Discussion
Line 282-283 "which is partly consistent with our results." Please explain in detail the word "partly" here.
Line 286-287 "During infection or injury, HMGB1 also functions outside the nucleus." Please add a reference to this sentence.
It is suggested to add inter-dose effect relationships and to explain why in relation to the corresponding indicators in the discussion section.
Reviewer 2 Report
Although the manuscript is interesting, extensive editing of the English language and style is required. In addition, the authors should consider some suggested points to improve the quality of the manuscript, as well as answer some questions before its publication.
Abstract
L.23. Include the meaning of APEC since it is the first time it is mentioned.
L.23. Luteolin, in the form of a glycosylated flavone,...
Introduction
L.39. Is one of the principals, what? Please rewrite to make it clearer.
L.63. Luteolin, in the form of a glycosylated flavone,...
Materials and Methods
L.85. Sima aldridge or sigma aldrich??? Include the catalog number, please.
L.86. divided into 4 groups and ... Delete "and" in the sentence.
L.87. The experimental groups,...
L.89-90. Delete, this sentece was mentioned before.
L.91. Change "were given" to "received".
L.92-93. Why was the challenge with E coli administered intramuscularly and not orally?
L.94. Superscript. It is important to describe how the bacterial culture was carried out until reaching the dose that was administered to each of the chickens.
L.95. Is the model used prophylactic? How would Luteolin work as a treatment, that is, first the infection process and then the administration of Luteolin? it has been done?
L.96. What was the method used to slaughter the chickens? Describe it please.
L.109. It seems that the sentence is not complete. Please, check it.
L.116. Change "washed by PBS" to "washed with PBS"
L.116-117. Rewrite this sentence to make it clearer.
L.121. were measured with a commercial kit...
L.123-126. Rewrite this sentence to make it clearer.
The wording is supposed to be in the past since it is a methodology that has already been carried out.
L.136. Include the meaning of RIPA.
L.139. Include the meaning of BCA.
L.139-140. The sentence is not understood and must be in the past tense. Please rewrite it.
L.140-142. Rewrite the sentence to make it clearer.
L.143. bodies were anti-HMGB1.
Statistical Analysis
What data was analyzed with Excel, please be more explicit?
Results
L.153. The body weight changes of chicks are presented in Table 3 and Figure. 1A.
L.155. Please reconsider rewriting the sentence.
L.157. Please reconsider rewriting the sentence.
L.172. in the L10 and L20 groups...
L.180. compared with the control group...
L.199-200. Rewrite the sentence in order to be clearer.
L.206. Change "As well as" to "Furthermore o In addition"
L.233. This sentence is repetitive since the same information is presented a few lines below. It is better to delete the sentence.
L.240. This sentence is confusing since it refers to The methodology that was followed to measure the levels of HMGB1, p-p65, TLR4 and p65, and NOT the results obtained from the expression levels. Please rewrite it.
L.243. In contrast to E. coli group, the protein expression.
L246. Delete "to".
Discussion
L.267-268. Include the reference.
L.287. Extracellular HMGB1 released.
L.311-312. Rewrite the sentence to make it clearer.
Figure 6. Why is the significance lower in the L20 group compared to L10 if the opposite is observed?
Reviewer 3 Report
In summary, atuhors coclude that that luteolin relieves oxidative stress, inflammation and 320 ileal tissue damage in chicks caused by APEC. However, they did not analyse luteolin or other phenolic compounds in the feed. The pure substances, if not natural may be not as effective as natural ones from plant extrcats. Why authors did not have a group of chickens with natural origin of the palant extract?
Analysis with kits are not so precise as with chemical compounds and chromatogram. Why they did not use another technique for MDA?
Number of chickens is very low. What was the statistical power of the trial?
This study describes the HMGB1/TLR4/NF- 321 κB signal axis, which is a new signaling axis that may be involved in APEC-induced in- 322 flammation, and helps us better understand the mechanism of luteolin in treating inflam- 323 matory diseases aroused by bacteria.
Did the authors analyse intestinal microbiota?
did the authors compare with natural origin substnaces?
Reviewer 4 Report
Cao et al. report here a well-conducted study that found luteolin attenuates APEC-induced oxidative stress and inflammation via inhibiting the HMGB1/TLR4/NF-κB signaling pathway in the ileum of chicks. The topic is interesting. The study is compactly and well written, the figures and tables are well-presented. This reviewer, however, has a few suggestions that would improve the manuscript for readers.
1. Line 23, please add the full name of APEC, which was firstly appeared in the paper.
2. Line 26, please add the details of the experimental designs, including replicates for each treatment.
3. Line 84, please add the approved information of the Animal Experimentation Ethics Committee.
4. Line 86, what is the species?
5. Line 158, 15 birds might not enough for the growth performance data. As chicks are small animals, the sample size needs to be improved.
6. Line 98-99, please change to expressed as kg diets containing how many micronutrients.
7. Line 135, these information should just put in the supplemental file.
8. When the treatment > 2, the significant changes of the results should be indicated as different letters.
9. No need to use both Figure 4E and F, Figure 5A and B, Figure 6A and B at the same time. Please remove one of them.
10. Please check the references and make sure they are followed the journal style.
Round 2
Reviewer 2 Report
Please review the attached file for some suggestions made, which are marked in green.

Reviewer 3 Report
Authors have replied in a very extensive way. Manuscript preparation is adequate and organisation is correct.
Author Response
We are honored to have been recognized by the reviewers, and we want to once again thank them for their comprehensive examination of our manuscript and insightful comments.